# Factors associated with hepatocellular carcinoma occurrence after HCV eradication in patients without cirrhosis or with compensated cirrhosis

**Kazumichi Abe**[1,2]*, **Hiroto Wakabayashi**[3], **Haruo Nakayama**[4], **Tomohiro Suzuki**[5], **Masahito Kuroda**[6], **Naoe Yoshida**[7], **Jun Tojo**[8], **Atsuko Kogure**[9], **Tsuyoshi Rai**[10], **Hironobu Saito**[10], **Shinji Mukai**[11], **Masashi Fujita**[1], **Manabu Hayashi**[1], **Atsushi Takahashi**[1], **Hiromasa Ohira**[1]

1 Department of Gastroenterology, Fukushima Medical University School of Medicine, Fukushima, Japan, 2 Department of Internal Medicine, Hanawa Kosei Hospital, Higashishirakawa, Japan, 3 Department of Gastroenterology, Takeda General Hospital, Aizuwakamatsu, Japan, 4 Department of Gastroenterology, Iwaki City Medical Center, Iwaki, Japan, 5 Department of Gastroenterology, Fukushima Rosai Hospital, Iwaki, Japan, 6 Department of Gastroenterology, Fukushima Red Cross Hospital, Fukushima, Japan, 7 Department of Internal Medicine, Iwase General Hospital, Sukagawa, Japan, 8 Cosmos Clinic, Koriyama, Japan, 9 Department of Gastroenterology, Fujita General Hospital, Date, Japan, 10 Rai Clinic, Miharu, Japan, 11 Department of Gastroenterology, Ohta Nishinouchi Hospital, Koriyama, Japan

* k-abe@fmu.ac.jp

**Data Availability Statement:** All relevant data are within the manuscript and its Supporting information files.

## Abstract

The present study aimed to investigate the incidence of hepatocellular carcinoma (HCC) and factors related to HCC occurrence after direct-acting antiviral (DAA) treatment in the Fukushima Liver Academic Group (FLAG). We conducted a multicenter retrospective cohort study of 1068 patients without cirrhosis (NC) or with compensated liver cirrhosis (LC) who achieved a sustained virologic response (SVR). First, we compared the cumulative HCC incidence and survival rates in NC (n = 880) and LC (n = 188) patients without a history of HCC treatment. Second, we performed multivariate analysis of factors related to HCC occurrence after DAA treatment. Overall, the average age was 65 years, and the male/female ratio was 511/557. Thirty-nine (4%) patients developed HCC. The cumulative 4-year HCC incidence and survival rates were 3.0% and 99.8% in NC patients and 11.5% and 98.5% in LC patients, respectively. The independent factors affecting HCC occurrence identified by multivariate analysis were the serum albumin (ALB) level before SVR for NC patients and the ALBI score, platelet count, and diabetes before SVR for LC patients. The factors related to HCC occurrence differed between NC and LC patients. Careful surveillance of post-SVR patients with these risk factors is needed.

## Introduction

Eradication of hepatitis C virus (HCV) by successful antiviral treatment with direct-acting antivirals (DAAs) has been shown to significantly reduce the risk of hepatocellular carcinoma

**Funding:** Rai Clinic provided support in the form of salaries for authors TR and HS. Cosmos Clinic provided support in the form of salary for author JT. Funders did not have any additional role in the study design, data collection and analysis, decision to publish, or preparation of the manuscript. The specific roles of these authors are articulated in the "author contributions" section.

**Competing interests:** Rai Clinic provided support in the form of salaries for authors TR and HS. Cosmos Clinic provided support in the form of salary for author JT. This does not alter our adherence to PLOS ONE policies on sharing data and materials. There are no patents, products in development or marketed products to declare.

(HCC) [1–5]. However, it is unclear whether the risk of HCC will decrease over time after the eradication of HCV. The absolute risk of HCC persists in patients with established cirrhosis prior to antiviral treatment, even after achieving sustained virologic response (SVR), despite a significantly reduced risk of HCC compared to that in untreated patients and those who did not achieve SVR [6]. Moreover, predictors of HCC occurrence after DAA treatment in patients without a history of HCC treatment have not been clearly elucidated. Previous studies have demonstrated that factors evaluated at or shortly before the onset of antiviral treatment and at the end of antiviral treatment can be used to predict the occurrence of HCC after the initiation of treatment. A study reported that pretreatment data are the most clinically useful data because clinical tests are routinely performed at the beginning of treatment, and treatment strongly influences the outcomes of many tests [6].

In the present study, we focused on using pretreatment data to evaluate the factors associated with HCC occurrence in post-SVR patients without cirrhosis or with compensated cirrhosis.

## Results

### Patient characteristics

The selection of the study population is illustrated in Fig 1. We identified 1263 HCV patients who underwent DAA treatment. We excluded 64 patients with missing data before and after DAA initiation, 91 patients who had a history of HCC treatment, and 40 patients with non-SVR status after DAA initiation. A total of 1068 remaining patients were included in the final analysis, of whom 880 did not have cirrhosis (NC) and 188 had compensated liver cirrhosis (LC). The patients' baseline characteristics are shown in Table 1. A total of 511 male and 557 female patients were enrolled. The median patient age at the start of treatment was 65 years. All patients achieved an SVR with DAA treatment. A history of IFN-based therapy (28%), the presence of diabetes mellitus (DM) (19%), and positivity for HBc antibody (6%) were identified. The mean observation period was 43 months after the initiation of DAA treatment.

Compared with NC patients, patients with LC were older (68 vs 64 years). Compared with NC patients, patients with LC had significantly higher levels of TB, AST, ALT, and γ-GTP (GGT); higher FIB-4 and ALBI scores; higher AFP concentrations; and significantly lower ALB levels and PLT counts.

### Cumulative HCC incidence and survival rates

The 1-, 2-, 3-, and 4-year incidence rates of HCC were 0.7%, 1.1%, 1.8%, and 3.0% for NC patients compared to 2.6%, 4.9%, 9.3%, and 11.5%, respectively, for LC patients (Fig 2a). The rates of HCC incidence for LC patients were significantly increased compared with those for NC patients ($P<0.0001$). Among NC patients, 12 deaths occurred during the study period, whereas among LC patients, 7 deaths occurred. In the 880 NC patients with SVR, 2 patients died due to liver-related causes (HCC and hepatic ascites), and 10 patients died due to non-liver-related causes during the observation period. Of the 10 patients with unrelated deaths, 2 patients died due to cholangiocarcinoma, 1 patient died due to gallbladder carcinoma, 1 patient died due to pancreatic carcinoma, 1 patient died due to renal cell carcinoma, 1 patient died due to malignant lymphoma, 1 patient died due to aortic dissection, and 1 patient died due to nontuberculous mycobacterial infection. In 2 patients, the cause of the death was unknown. In 188 LC patients with SVR, 2 patients died due to HCC, 1 patient died due to liver failure, 1 patient died due to liver abscess, and 3 patients died due to non-liver-related causes during the observation period. Of the 3 patients with unrelated deaths, 1 patient died due to cerebral hemorrhage, and 2 patients died due to unknown cause. The 1-, 2-, 3-, and 4-year

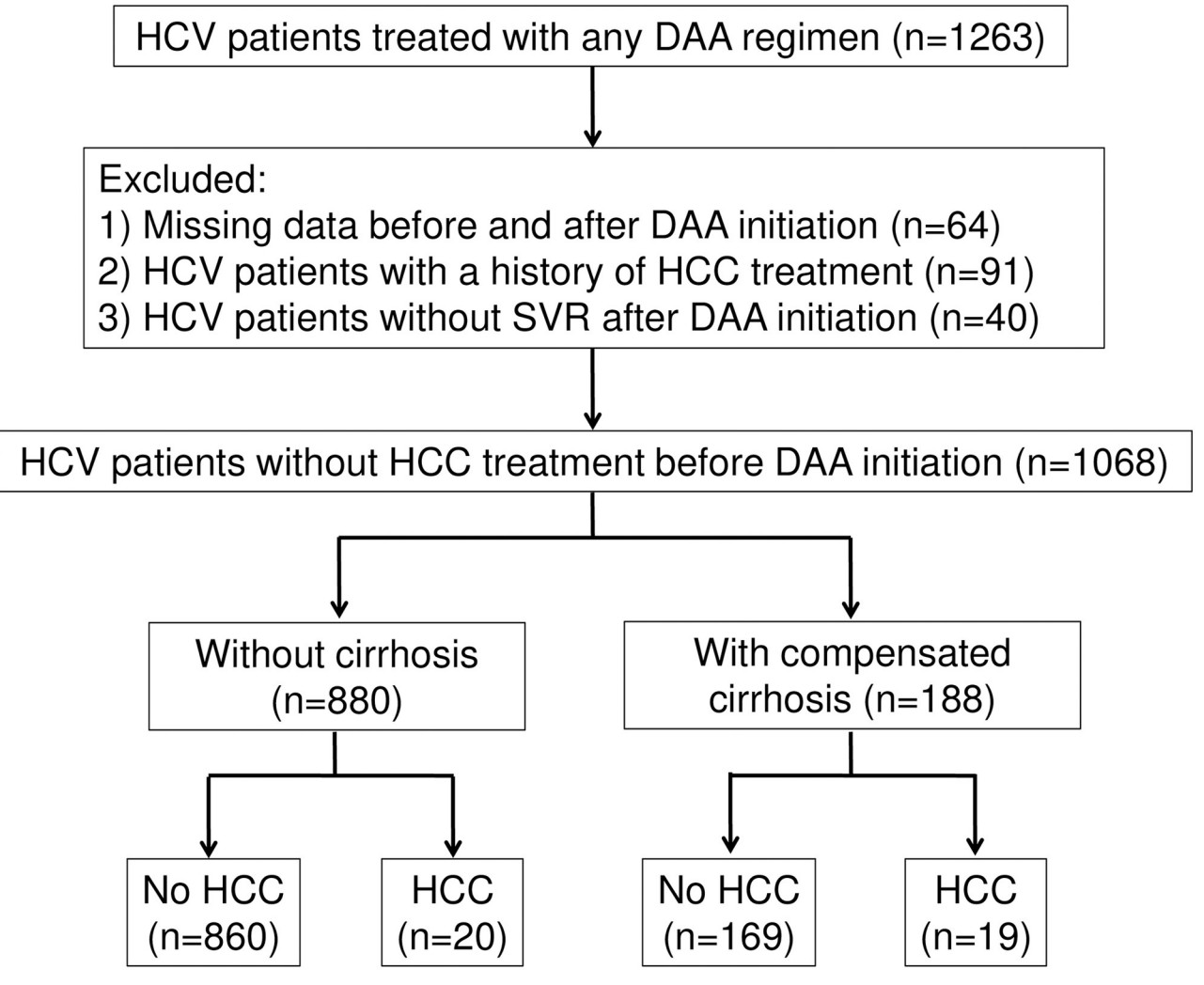

**Fig 1. Diagram of the patient selection process.**

liver-related survival rates were 99.9%, 99.8%, 99.8%, and 99.8%, respectively, for NC patients compared to 99.5%, 99.5%, 99.5%, and 98.5%, respectively, for LC patients (Fig 2b). The liver-related survival rates for LC patients with SVR were significantly poorer than those for NC patients with SVR ($P<0.01$). The all survival rates and non-liver-related survival rates were not significantly different between NC patients and LC patients after HCV eradication (S1 Fig). Moreover, the achievement of SVR that potentially contributed to HCC development after DAA treatment in 1108 HCV-positive patients, included 40 patients without SVR, was also evaluated (S1–S3 Tables, S2 Fig). The proportion of patients with SVR was significantly reduced in LC patients compared with NC patients (90% vs 98%, $P<0.0001$). Among NC patients but not LC patients, the proportion of patients with SVR was significantly lower in patients with HCC than in those without HCC (91% vs 98%, $P<0.05$).

### Predictors of HCC occurrence after DAA treatment

Pretreatment factors that might have contributed to HCC development after DAA treatment in NC patients were evaluated (Table 2). Among NC patients (n = 880), patients who

**Table 1. Characteristics of 1068 patients with hepatitis C virus.**

| | All (n = 1068) | No cirrhosis (n = 880) | Compensated cirrhosis (n = 188) | P value |
|---|---|---|---|---|
| Age, years, median (IQR) | 67 (57–75) | 66 (56–74) | 70 (61–77) | <0.0001* |
| Sex, male/female (male %) | 511/557 (48%) | 421/459 (48%) | 90/98 (48%) | 0.3546 |
| Genotype, 1/2/other (G-1%) | 720/341/7 (67%) | 589/285/6 (67%) | 131/56/1 (70%) | 0.4567 |
| HCV RNA, LogIU/ml, median (IQR) | 6.2 (5.6–6.6) | 6.2 (5.7–6.6) | 6.1 (5.5–6.5) | 0.0788 |
| History of interferon-based therapy, yes (%) | 303 (28%) | 244 (28%) | 59 (31%) | 0.3128 |
| Diabetes mellitus, n (%) | 198 (19%) | 154 (18%) | 44 (23%) | 0.0586 |
| HBcAb-positive, n (%) | 60 (6%) | 51 (6%) | 9 (5%) | 0.5858 |
| Observation period after DAA treatment, months, median (IQR) | 43 (32–49) | 42 (31–48) | 46 (37–52) | <0.0001* |
| ALB, g/dl, median (IQR) | 4.1 (3.8–4.4) | 4.2 (3.9–4.4) | 3.9 (3.6–4.1) | <0.0001* |
| TB, mg/dl, median (IQR) | 0.8 (0.6–1.0) | 0.8 (0.6–0.9) | 0.8 (0.7–1.1) | <0.0001* |
| AST, U/l, median (IQR) | 40 (27–60) | 37 (26–53) | 53 (39–76) | <0.0001* |
| ALT, U/l, median (IQR) | 38 (26–62) | 36 (25–60) | 47 (32–76) | <0.0001* |
| GGT, U/l, median (IQR) | 32 (20–57) | 31 (19–57) | 39 (24–60) | 0.0052* |
| eGFR, ml/min/1.73 m$^2$, median (IQR) | 72 (62–83) | 72 (62–83) | 71 (59–83) | 0.2938 |
| PLT, ×10$^4$/μl, median (IQR) | 15.7 (12.0–20.0) | 17.0 (13.6–20.9) | 9.0 (7.1–11.1) | <0.0001* |
| FIB-4 score, median (IQR) | 2.8 (1.8–4.4) | 2.4 (1.6–3.6) | 6.2 (4.3–8.7) | <0.0001* |
| ALBI score, median (IQR) | -2.8 (-3.0- -2.5) | -2.8 (-3.0- -2.6) | -2.5 (-2.8- -2.2) | <0.0001* |
| AFP, ng/ml, median (IQR) | 4.0 (2.6–6.9) | 3.6 (2.5–5.8) | 7.8 (4.2–20.2) | <0.0001* |

*P < 0.05 was considered significant (no cirrhosis vs compensated cirrhosis).

Abbreviations: DAA, direct-acting antiviral; HCC, hepatocellular carcinoma; ALB, albumin; TB, total bilirubin; AST, aspartate aminotransferase; ALT, alanine aminotransferase; GGT, γ-glutamyltransferase; PLT, platelet count; FIB-4, fibrosis-4; ALBI, albumin–bilirubin; AFP, α-fetoprotein; IQR, interquartile range.

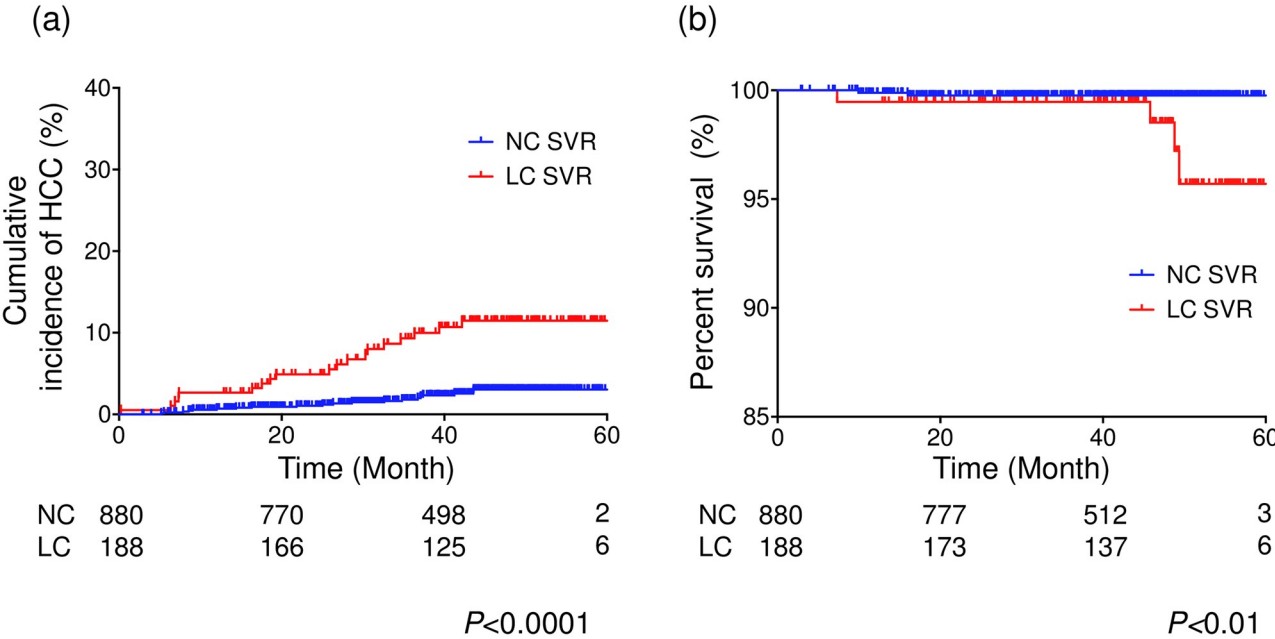

**Fig 2. Comparison of the cumulative hepatocellular carcinoma incidence and liver-related survival rate between hepatitis C patients without cirrhosis (NC) and those with compensated liver cirrhosis (LC).** (a) Cumulative incidence of hepatocellular carcinoma in patients with compensated liver cirrhosis (red line) and patients without cirrhosis (blue line). (b) The liver-related survival rate of patients with compensated liver cirrhosis (red line) and patients without cirrhosis (blue line). The results were analyzed with the log-rank test.

**Table 2. Pretreatment factors associated with the development of hepatocellular carcinoma after DAA treatment in HCV-positive patients without cirrhosis.**

| | No HCC (n = 860) | HCC (n = 20) | P value |
|---|---|---|---|
| Age, years, median (IQR) | 66 (56–74) | 73 (68–75) | 0.0059* |
| Sex, male/female (male %) | 407/453 (47%) | 14/6 (70%) | 0.0448* |
| Genotype, 1/2/other (G-1%) | 572/282/6 (67%) | 17/3/0 (85%) | 0.0816 |
| HCV RNA, LogIU/ml, median (IQR) | 6.2 (5.7–6.6) | 6.0 (5.5–6.2) | 0.0249* |
| History of interferon-based therapy, yes (%) | 236 (27%) | 8 (40%) | 0.2149 |
| Diabetes mellitus, n (%) | 144 (17%) | 10 (50%) | 0.0001* |
| HBcAb-positive, n (%) | 48 (6%) | 3 (15%) | 0.0747 |
| Observation period after DAA treatment, months, median (IQR) | 42 (31–48) | 46 (41–49) | 0.1034 |
| ALB, g/dl, median (IQR) | 4.2 (3.9–4.4) | 3.9 (3.6–4.2) | 0.002* |
| TB, mg/dl, median (IQR) | 0.8 (0.6–0.9) | 0.8 (0.6–1.0) | 0.9518 |
| AST, U/l, median (IQR) | 36 (26–53) | 51 (44–59) | 0.0396* |
| ALT, U/l, median (IQR) | 36 (25–59) | 46 (36–60) | 0.2099 |
| GGT, U/l, median (IQR) | 31 (19–56) | 40 (20–55) | 0.7891 |
| eGFR, ml/min/1.73 m$^2$, median (IQR) | 72 (62–83) | 78 (73–84) | 0.1688 |
| PLT, ×10$^4$/µl, median (IQR) | 17.0 (13.9–21.0) | 13.0 (10.8–16.8) | 0.0004* |
| FIB-4 score, median (IQR) | 2.4 (1.6–3.6) | 4.2 (3.0–5.2) | <0.0001* |
| ALBI score, median (IQR) | -2.8 (-3.0- -2.6) | -2.6 (-2.9- -2.4) | 0.0046* |
| AFP, ng/ml, median (IQR) | 3.5 (2.5–5.7) | 6.2 (3.7–7.4) | 0.006* |

*$P < 0.05$ was considered significant (no HCC vs HCC).

Abbreviations: DAA, direct-acting antiviral; HCC, hepatocellular carcinoma; ALB, albumin; TB, total bilirubin; AST, aspartate aminotransferase; ALT, alanine aminotransferase; GGT, γ-glutamyltransferase; PLT, platelet count; FIB-4, fibrosis-4; ALBI, albumin–bilirubin; AFP, α-fetoprotein; IQR, interquartile range.

developed HCC (n = 20) were older (72 vs 64 years) and more likely to be male (70% vs 40%) than patients who did not develop HCC (n = 860). The proportion of patients with DM was significantly higher in patients with HCC than in those without HCC (50% vs 17%). NC patients who developed HCC had higher AST levels, FIB-4 scores, ALBI scores, and AFP concentrations and lower ALB levels and PLT counts before DAA treatment than those who did not develop HCC. The results of multivariate Cox hazard regression analysis with the factors significantly associated with HCC occurrence in NC patients are shown in Table 3. According to the ROC curve for HCC occurrence based on the ALB level noted in 880 patients without cirrhosis (sensitivity, 58.82%; specificity, 74.11%; AUC, 0.7145), the best cutoff value for the ALB level before DAA treatment was 3.95 g/dl. After adjustment for confounding variables, such as age and sex, the hazard ratio (HR) of HCC occurrence with a serum ALB level <3.95 g/dl was 3.57 (95% CI, 1.35–9.95). After adjustment for confounding variables, such as age, sex, AFP level, FIB-4 score, and the presence of DM, the HR of HCC occurrence with a serum ALB level <3.95 g/dl was 3.62 (95% CI, 1.22–12.11). A serum ALB level <3.95 g/dl was an independent risk factor for HCC occurrence in NC patients. As assessed by the log-rank test, the cumulative HCC incidence was significantly higher in NC patients whose serum ALB level was <3.95 g/dl than in NC patients whose serum ALB level was ≥3.95 g/dl (P = 0.0013, S3a Fig). Similarly, the cumulative HCC incidence was significantly higher in NC patients whose serum AFP level was >6 ng/ml than in NC patients whose serum AFP level was ≤6 ng/ml (P = 0.0035, S3b Fig). Furthermore, the cumulative HCC incidence was significantly higher in NC patients who had DM than in NC patients who did not have DM (P<0.0001; S3c Fig). Similarly, the cumulative HCC incidence was significantly higher in NC patients whose FIB-4

**Table 3. Factors associated with the development of HCC after DAA treatment in HCV-positive patients without cirrhosis by Cox multivariate model.**

| HCC vs No HCC | | Model 1[a] | | Model 2[b] | |
|---|---|---|---|---|---|
| | | HR (95% CI) | *P* | HR (95% CI) | *P* |
| ALB (g/dl) | <3.95 | 3.57 (1.35–9.95) | 0.0104* | 3.62 (1.22–12.11) | 0.0198* |
| | ≥3.95 | 1 (Ref) | | 1 (Ref) | |
| AFP (ng/ml) | >6 | | | 2.20 (0.70–7.21) | 0.1768 |
| | ≤6 | | | 1 (Ref) | |
| DM | Yes | | | 2.14 (0.68–6.36) | 0.1871 |
| | No | | | 1 (Ref) | |
| FIB-4 score | ≥3.25 | | | 1.84 (0.53–7.01) | 0.3413 |
| | <3.25 | | | 1 (Ref) | |

Ref, reference group; HR, hazard ratio; CI, confidence interval.

[a]After adjusting for age and sex.

[b]After adjusting for age, sex, AFP, DM and FIB-4 score.

*$P < 0.05$ was considered significant.

Abbreviations: HCC, hepatocellular carcinoma; AFP, α-fetoprotein; ALB, albumin; DM, diabetes mellitus; FIB-4, fibrosis-4.

score was ≥3.25 than in NC patients whose FIB-4 score was <3.25 (*P* = 0.0008, S3d Fig). Post-treatment factors that might contribute to HCC development after the end of DAA treatment were also evaluated. Multivariate analysis identified FIB-4 score ≥3.25 and AFP level >6 ng/ml after DAA treatment as independent factors that contributed to the development of HCC in patients without cirrhosis (S5 and S6 Tables).

On the other hand, among LC patients (n = 188), those who developed HCC (n = 19) were more likely to have DM, be infected with genotype 1, and have a history of IFN-based therapy than those who did not develop HCC (n = 169) (Table 4). LC patients with HCC had higher TB levels and ALBI scores and lower ALB levels and PLT counts before DAA treatment than those who did not develop HCC. The results of multivariate logistic regression analysis with the factors significantly associated with HCC occurrence in LC patients are shown in Table 5. According to the ROC curve for HCC occurrence based on the PLT counts in 188 patients with compensated cirrhosis (sensitivity, 63.16%; specificity, 65.09%; AUC, 0.6434), the best cutoff value for the PLT count before DAA treatment was 8.2 x$10^4$/μl. Furthermore, the cutoff value for the ALBI score before DAA treatment was -2.3 according to the ROC curve for HCC occurrence based on the ALBI score in 188 patients with compensated cirrhosis (sensitivity, 57.89%; specificity, 74.53%; AUC, 0.7310). After adjustment for confounding variables, such as age and sex, the HR for HCC occurrence with an ALBI score >-2.3 versus an ALBI score ≤-2.3 was 4.38 (95% CI, 1.77–11.36). After adjustment for confounding variables, such as age, sex, DM status, and PLT count, the HR for the development of HCC with an ALBI score >-2.3 was 4.26 (95% CI, 1.70–11.15), the HR with the presence of DM was 3.80 (95% CI, 1.35–10.65), and the HR with a PLT count <8.2 ×$10^4$/μl was 4.14 (95% CI, 1.55–11.20). The ALBI score, DM status and PLT count were independent risk factors for the development of HCC in LC patients. Posttreatment factors that might contribute to HCC development after the end of DAA treatment were also evaluated. Multivariate analysis identified an ALBI score >-2.3 after DAA treatment as an independent factor that contributed to the development of HCC in patients with compensated cirrhosis (S7 and S8 Tables).

## Subgroup analysis of HCC occurrence

The incidence of HCC was examined by including the factors identified as contributing to HCC occurrence after DAA treatment in multivariate analysis of pretreatment factors, namely,

**Table 4. Pretreatment factors associated with the development of HCC after DAA treatment in HCV-positive patients with compensated liver cirrhosis.**

|  | No HCC | HCC | P value |
|---|---|---|---|
|  | (n = 169) | (n = 19) |  |
| Age, years, median (IQR) | 71 (60–77) | 70 (68–75) | 0.8073 |
| Sex, male/female (male %) | 78/91 | 12/7 | 0.1595 |
|  | (46%) | (63%) |  |
| Genotype, 1/2/other (G-1%) | 114/54/1 | 17/2/0 | 0.0483* |
|  | (67%) | (89%) |  |
| HCV RNA, LogIU/ml, median (IQR) | 6.1 (5.5–6.6) | 6.1 (5.6–6.3) | 0.5160 |
| History of interferon-based therapy, yes (%) | 49 | 10 | 0.0353* |
|  | (29%) | (53%) |  |
| Diabetes mellitus, n (%) | 35 (21%) | 9 (47%) | 0.0093* |
| HBcAb-positive, n (%) | 7 (4%) | 2 (11%) | 0.2165 |
| Observation period after DAA treatment, months, median (IQR) | 46 (37–52) | 47 (35–55) | 0.8521 |
| ALB, g/dl, median (IQR) | 3.9 (3.6–4.1) | 3.6 (3.3–3.9) | 0.003* |
| TB, mg/dl, median (IQR) | 0.8 (0.6–1.1) | 1.1 (0.8–1.3) | 0.0155* |
| AST, U/l, median (IQR) | 53 (39–77) | 52 (37–62) | 0.6454 |
| ALT, U/l, median (IQR) | 45 (32–78) | 50 (30–67) | 0.8991 |
| GGT, U/l, median (IQR) | 39 (25–60) | 32 (22–63) | 0.6742 |
| eGFR, ml/min/1.73 m$^2$, median (IQR) | 71 (61–83) | 68 (57–82) | 0.9894 |
| PLT, ×10$^4$/μl, median (IQR) | 9.2 (7.1–11.7) | 8.1 (6.6–9.1) | 0.04* |
| FIB-4 score, median (IQR) | 6.1 (4.2–8.7) | 7.4 (5.3–8.9) | 0.2425 |
| ALBI score, median (IQR) | -2.6 (-2.8- -2.3) | -2.2 (-2.5- -1.9) | 0.0008* |
| AFP, ng/ml, median (IQR) | 7.6 (4.1–20.2) | 15.0 (6.5–21.9) | 0.1387 |

*$P < 0.05$ was considered significant (no HCC vs HCC).

Abbreviations: DAA, direct-acting antiviral; HCC, hepatocellular carcinoma; ALB, albumin; TB, total bilirubin; AST, aspartate aminotransferase; ALT, alanine aminotransferase; GGT, γ-glutamyltransferase; PLT, platelet count; FIB-4, fibrosis-4; ALBI, albumin–bilirubin; AFP, α-fetoprotein; IQR, interquartile range.

**Table 5. Factors associated with the development of HCC after DAA treatment in HCV-positive patients with compensated liver cirrhosis by Cox multivariate model.**

| HCC versus No HCC |  | Model 1[a] | | Model 2[b] | |
|---|---|---|---|---|---|
|  |  | HR (95% CI) | P | HR (95% CI) | P |
| ALBI score | >-2.3 | 4.38 (1.77–11.36) | 0.0016* | 4.26 (1.70–11.15) | 0.0022* |
|  | ≤-2.3 | 1 (Ref) |  | 1 (Ref) |  |
| DM | Yes |  |  | 3.80 (1.35–10.65) | 0.0119* |
|  | No |  |  | 1 (Ref) |  |
| PLT (×10$^4$/μl) | <8.2 |  |  | 4.14 (1.55–11.20) | 0.0046* |
|  | ≥8.2 |  |  | 1 (Ref) |  |

Ref, reference group; HR, hazard ratio; CI, confidence interval.

[a]After adjusting for age and sex.

[b]After adjusting for age, sex, DM, and PLT.

*$P < 0.05$ was considered significant.

Abbreviations: HCC, hepatocellular carcinoma; ALBI, albumin–bilirubin; DM, diabetes mellitus; PLT, platelet count.

## Compensated cirrhosis group

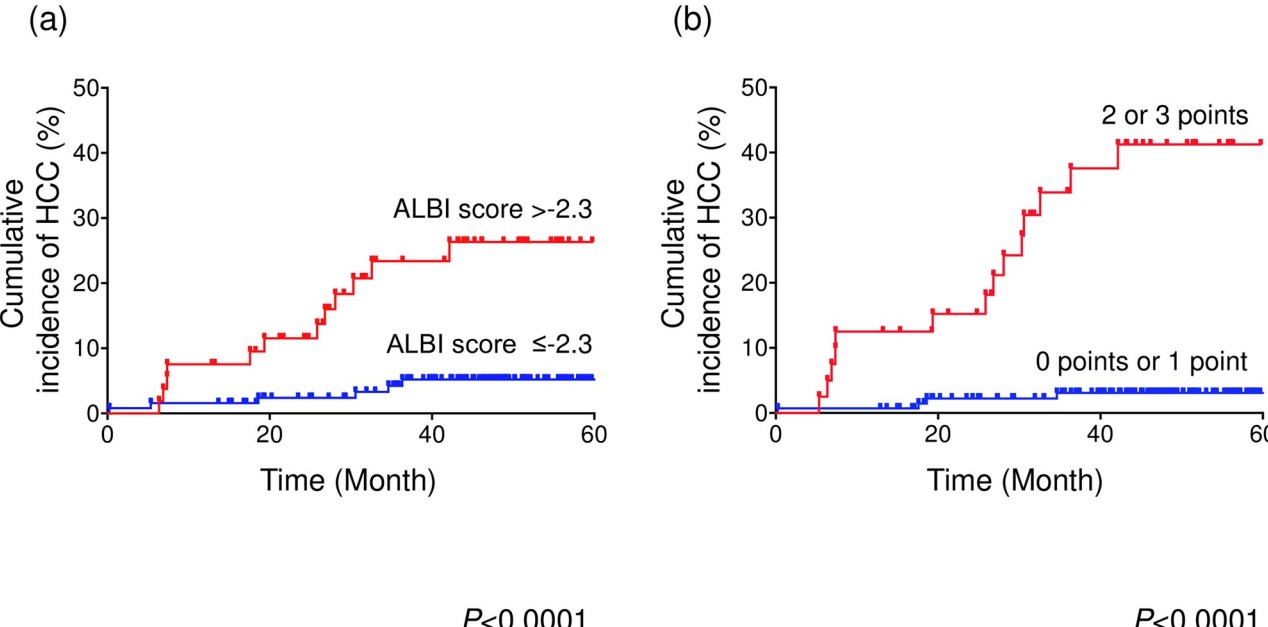

$P<0.0001$ $P<0.0001$

**Fig 3. Comparison of the cumulative HCC incidence between compensated liver cirrhosis patients with HCV grouped according to factors that might contribute to the development of HCC after DAA treatment.** (a) ALBI score >-2.3 (red line); ALBI score ≤-2.3 (blue line). (b) Score combining the ALBI score, PLT count, and presence of DM that was 2 or 3 points (red line) or 0 points or 1 point (blue line). The results were analyzed with the log-rank test.

the DM status, ALBI score, and PLT count. As determined by the log-rank test, the cumulative HCC incidence was significantly higher in LC patients whose ALBI score was >-2.3 than in LC patients whose ALBI score was ≤-2.3 ($P<0.0001$; Fig 3a). The 1-, 2-, 3-, and 4-year incidence rates of HCC for LC patients whose ALBI score was ≤-2.3 were 1.6%, 2.4%, 4.2%, and 5.2%, compared to 7.5%, 11.5%, 23.4%, and 26.3%, respectively, among LC patients whose ALBI score was >-2.3. Similarly, the cumulative HCC incidence was significantly higher in LC patients whose PLT count was $<8.2 \times 10^4$/μl than in LC patients whose PLT count was $\geq 8.2 \times 10^4$/μl ($P<0.05$, S4a Fig). Additionally, the cumulative HCC incidence was significantly higher in LC patients with DM than in LC patients without DM ($P<0.005$, S4b Fig).

### Calculation of the score combining the ALBI score, PLT count, and presence of DM

Based on the results of the multivariate analysis that included pretreatment factors, a scoring system that combined the ALBI score, PLT count, and DM status was developed. ALBI scores ≤-2.3 and >-2.3 were scored as 0 and 1, respectively. Pretreatment PLT ($\times 10^4$/μl) counts ≥8.2 and <8.2 were scored as 0 and 1, respectively. The absence of DM and the presence of DM were scored as 0 and 1, respectively.

### HCC occurrence stratified by the score combining the ALBI score, PLT count, and presence of DM

Of the study patients, 141 had scores of 0 or 1, and 40 had scores or 2 or 3. The study patients were then grouped based on these scores as follows: those with 0 points or 1 point constituted

the low-score group, while those with 2 or 3 points were categorized in the high-score group. Fig 3b shows the cumulative incidence of HCC for the two groups. As determined by the log-rank test, the cumulative HCC incidence was significantly higher in the high-score group than in the low-score group ($P$<0.0001). The 1-, 2-, 3-, and 4-year incidence rates of HCC in the low-score group were 0.7%, 2.2%, 3.1%, and 3.1% compared to 12.5%, 15.2%, 33.9%, and 41.2%, respectively, in the high-score group.

## Discussion

Most patients with HCV in Japan have undergone DAA-based treatment, and the vast majority will achieve an SVR. Furthermore, an SVR is associated with improved clinical outcomes, including a reduced risk of HCC [1–5]. Several studies have analyzed the risk factors for HCC occurrence and the characteristics of HCC that occurs after SVR in chronic hepatitis C patients treated with DAA compared with those treated with IFN-based treatments. The AFP and ALT levels after IFN-based treatment are previously reported risk factors associated with HCC occurrence [7]. Other studies have reported that the risk factors for HCC occurrence after IFN-based treatment are advanced age, advanced fibrosis, fatty liver, male sex and alcohol consumption [8–10]. On the other hand, another study demonstrated that in patients without cirrhosis, higher FIB-4 scores, the presence of DM, and alcohol use were risk factors for developing HCC after DAA treatment [3]. In addition, recent studies reported that male sex, increased FIB-4 scores, decreased ALB levels, older age, and increased AFP levels were independent risk factors identified by multivariate analysis [11–13]. In the present study, a long-term observation period of more than 3 years was used, and the HCC risk factors after HCV eradication were identified in patients with compensated cirrhosis or without cirrhosis. The 1-, 2-, 3-, and 4-year incidence rates of HCC for patients without cirrhosis were 0.7%, 1.1%, 1.8%, and 3.0%, respectively. Furthermore, multivariate analysis identified a serum ALB level <3.95 before DAA treatment as an independent factor that contributed to the development of HCC in patients without cirrhosis. The proportion of patients with DM was significantly increased in patients with HCC compared with those without HCC, and DM was extracted as a related factor for the incidence of HCC in univariate analysis (S4 Table). However, DM was not extracted by multivariate analysis. In the present study, NC patients presented a large amount of chronic hepatitis with fibrosis, and ALB was extracted by multivariate analysis rather than DM compared to the previously reported mild liver disease [14, 15].

On the other hand, cirrhotic patients who achieve an SVR represent a particularly difficult conundrum for health care providers. A 5-year long-term follow-up study after DAA treatment confirmed a reduced HCC risk in a central European cohort of HCV patients with cirrhosis, with HCC incidences of 2.04 (DAA-treated) and 5.04 (untreated) per 100 person-years [16]. Overall, 5.7% (DAA-treated) vs 12.5% (untreated) of cirrhotic patients developed HCC. Achieving SVR clearly reduces the risk of developing HCC, but patients with cirrhosis still have an absolute risk of developing HCC, which is worth monitoring [6]. However, analysis of a retrospective cohort study of 22,500 patients treated with DAAs in 129 Veterans Health Administration hospitals in 2015 reported an annual incidence of HCC after SVR of 1.82 per 100 person-years in patients with a diagnosis of cirrhosis [3]. In the present study, post-SVR patients with compensated cirrhosis were older and had a significantly higher incidence of HCC and mortality rate than those without cirrhosis. The 1-, 2-, 3-, and 4-year incidence rates of HCC for patients with compensated cirrhosis were 2.6%, 4.9%, 9.3%, and 11.5%, respectively. Furthermore, among patients with compensated cirrhosis, those who developed HCC had higher rates of DM, higher ALBI scores, and lower PLT counts before DAA treatment than those who did not develop HCC.

A new assessment tool for hepatic function, the ALBI score, which consists of only ALB and TB measurements, has recently been proposed [17, 18]. The ALBI score may be used to evaluate liver function damage and the prognosis of patients with liver cancer [19, 20]. This score has been reported to have predictive value for mortality in patients with both cirrhosis and upper gastrointestinal bleeding [21]. Only one previous report demonstrated that for cirrhotic patients with no history of HCC, an increase in the ALBI grade was independently associated with HCC development in multivariate analysis [22]. In the present study, multivariate analysis identified an ALBI score greater than -2.3 as an independent factor associated with the development of HCC in LC patients. Moreover, the incidence of HCC was examined by including the factors that contributed to the development of HCC after DAA treatment in multivariate analysis with the ALBI score. As determined by the log-rank test, the cumulative HCC incidence was significantly higher in LC patients whose ALBI score was >-2.3 than in LC patients whose ALBI score was ≤-2.3.

Thrombocytopenia, defined as a PLT count $<15 \times 10^4$/μl, is one of the most common hematological abnormalities in patients with chronic liver disease [23]. Causes of thrombocytopenia in patients with HCV-induced chronic liver disease include the consequences of portal hypertension and hypersplenism secondary to splenomegaly. Other causes include myelosuppression by HCV, immune-mediated platelet destruction including antiplatelet antibody and/or immune complex-related platelet clearance; and impaired thrombopoietin production due to hepatocyte injury [23].

Furthermore, in patients with cirrhosis, severe thrombocytopenia has been identified as an independent risk factor for developing complications of cirrhosis and death [24]. A recent study of 1170 patients with HCV genotype 1-related chronic liver disease treated with DAAs demonstrated through multivariate analysis that thrombocytopenia was a risk factor for HCC [25]. Another study revealed an association between HCC and DAA treatment in patients with decompensated cirrhosis. In addition, thrombocytopenia and DM were found to increase the likelihood of HCC after DAA treatment in patients with decompensated cirrhosis [26]. In the present study, multivariate analysis identified a PLT count less than $8.2 \times 10^4$/μl as an independent factor associated with the development of HCC in patients with compensated cirrhosis. Furthermore, the cumulative HCC incidence was significantly higher in LC patients whose PLT count was $<8.2 \times 10^4$/μl than in LC patients whose PLT count was $\geq 8.2 \times 10^4$/μl.

Furthermore, DM and cirrhosis are generally considered to be strong risk factors for HCC development after achieving an SVR [3, 27]. A previous study demonstrated that 399 patients with advanced pretreatment fibrosis showed that DM is a major risk factor for the development of HCC after the achievement of an SVR. Patients with cirrhosis and DM had a risk of developing HCC that was 7 times higher than that of patients without DM, and the annual HCC risk for this group was 7.9% during the first 2 years after achieving an SVR [28]. In the present study, multivariate analysis identified DM as an independent factor contributing to HCC occurrence in patients with compensated cirrhosis, and the cumulative HCC incidence was significantly higher in cirrhotic patients with DM than in those without DM. Moreover, in the present study, a score combining the ALBI score, PLT count, and DM status was proposed to predict the risk of HCC occurrence after HCV eradication. ALBI score values ≤-2.3 and >-2.3 were scored as 0 and 1, respectively. Pretreatment PLT ($\times 10^4$/μl) counts ≥8.2 and <8.2 were scored as 0 and 1, respectively. The absence of DM and the presence of DM were scored as 0 and 1, respectively. The 1-, 2-, 3-, and 4-year incidence rates of HCC for the low-score group (0 points or 1 point) were 0.7%, 2.2%, 3.1%, and 3.1% compared to 12.5%, 15.2%, 33.9%, and 41.2%, respectively, in the high-score group (2 or 3 points). Previous studies reported the stratification of HCC incidence by FIB-4 scores and AFP levels after SVR and by scores composed of patient sex and age, with and without cirrhosis [11, 12]. On the other

hand, the score established in this study is useful for patients with compensated cirrhosis who still have an absolute risk of residual HCC.

Our study has several limitations. First, the sample population was relatively small. Second, our data were limited to a review of charts from our institutions. Third, whether the score established in the present study is useful for patients with decompensated cirrhosis needs to be verified. In addition, we used a retrospective design; thus, our results need confirmation in a prospective study. Further, we should validate the ability of these factors to predict HCC occurrence after HCV eradication in other cohorts.

In conclusion, our retrospective cohort study used a long-term observation period of more than 3 years and identified risk factors for the development of HCC after HCV eradication in patients with compensated cirrhosis (i.e., LC patients) or without cirrhosis (i.e., NC patients). The factors related to HCC occurrence differed between NC and LC patients. Thus, a useful score was established for patients with compensated cirrhosis. Post-SVR patients with these risk factors should be monitored carefully for the possible development of HCC.

## Methods

### Study design and patient population

We conducted a multicenter retrospective cohort study of adult patients with HCV-related HCC who were treated with a DAA regimen from December 2014 through April 2019. Liver cirrhosis was diagnosed based on morphologic changes in the liver, such as hypertrophy of the left lateral and caudate lobes or atrophy of the right posterior haptic lobe identified by ultrasonography (US), computed tomography (CT), and/or magnetic resonance imaging (MRI); a finding of pseudolobule formation on histopathologic examination [29]; or the presence of portal hypertension, indicated by varices.

A total of 1068 consecutive patients infected with HCV genotypes 1–4 who were treated with a DAA regimen at any of 11 hospitals belonging to the Fukushima Liver Academic Group (FLAG) in Japan (Fukushima Medical University, Takeda General Hospital, Iwaki City Medical Center, Fukushima Rosai Hospital, Fukushima Red Cross Hospital, Iwase General Hospital, Cosmos Clinic, Fujita General Hospital, Hanawa Kosei Hospital, Rai Clinic, and Ohta Nishinouchi Hospital) between 2014 and 2019 were reviewed. The treatments for HCV genotype 1 infection included a 24-week combined regimen of NS5A and NS3 protease-targeted DAAs (daclatasvir [DCV] and asunaprevir [ASV]) in 230 patients and a 12-week treatment with NS5A protease- and NS5B polymerase-targeted DAAs (sofosbuvir [SOF] and ledipasvir [LDV]) in 354 patients [11]; a 12-week combined regimen of NS5A and NS3 protease-targeted DAAs (elbasvir [EBR] and grazoprevir [GZR]) in 90 patients [30]; and ombitasvir/paritaprevir/ritonavir (OBV/PTR/r) in 15 patients [31]. The treatments for HCV genotype 2 infection included a 12-week combined regimen of SOF and ribavirin (RBV) in 302 patients. Two brands of RBV were given: Copegus (Chugai Pharmaceutical, Tokyo, Japan) and Rebetol (Merck Sharp & Dohme, Whitehouse Station, NJ, USA). Ribavirin was administered as an oral tablet (600–1000 mg total daily dose, depending on the body weight). The need for dose changes, temporary interruptions, or the discontinuation of RBV were determined based on the manufacturer's prescribing information [11]. A total of 77 patients were treated with glecaprevir/pibrentasvir (G/P). HCV genotype 1- or genotype 2-infected patients without cirrhosis were treated with G/P for 8 weeks, and HCV genotype 1- or genotype 2-infected patients with compensated cirrhosis were treated with G/P for 12 weeks. Patients with HCV genotype 3/4 infections were treated with G/P for 12 weeks [32].

The study protocol conformed to the ethics guidelines of the Declaration of Helsinki. The study protocol was reviewed and opt-out consent was approved by the ethics committee of

Fukushima Medical University (No. 29021). The need to obtain informed consent from the participants was waived by the ethics committee of Fukushima Medical University due to the retrospective nature of the study. The study was conducted in accordance with the approved guidelines.

## Sustained virologic response (SVR)

We defined an SVR as a serum HCV RNA viral load below the lower limit of detection performed at least 12 weeks after the end of HCV treatment [33].

## Clinical and laboratory assessments

Patients with severe liver dysfunction, such as decompensated cirrhosis or residual HCC, were excluded. Patients with severe renal impairment were not given the SOF-based regimen (estimated glomerular filtration rate [eGFR] $< 30$ ml/min/1.73 m$^2$) or RBV (eGFR $< 50$ ml/min/1.73 m$^2$) [34]. We sent a questionnaire to each facility and performed the analyses based on the collected data. Detailed clinical and demographic information, including patient age, sex, history of interferon (IFN)-based therapy, presence of diabetes mellitus (DM), positivity for HBc antibody, and fatty liver, was collected. Laboratory data, including the level of the tumor marker α-fetoprotein (AFP), the albumin–bilirubin (ALBI) score and fibrosis-4 (FIB-4) score, were evaluated at the time of DAA treatment and 24 weeks after DAA treatment. DM was defined by any of the following criteria: (i) a documented history of diabetes, (ii) the administration of a diabetes medication, or (iii) a fasting glucose level $\geq 126$ mg/dl or an HbA1c level $\geq 6.5$ on two separate occasions [35].

## FIB-4 score before treatment and after SVR

We used baseline laboratory test data obtained within 6 months prior to initiation of DAA treatment to calculate the baseline FIB-4 score as follows [36]:

$$(\text{Age} \times \text{AST})/(\text{Platelet count [PLT]} \times \sqrt{\text{ALT}})$$

## ALBI score before treatment and after SVR

Recently, a simple and objective method for evaluating hepatic reserve function using only albumin (ALB) and total bilirubin (TB) measurements was proposed. We used baseline laboratory test data obtained within 6 months before the initiation of DAA treatment to calculate the baseline ALBI score as follows [17]:

$$\text{Albumin} - \text{bilirubin (ALBI) grade } ([\log 10 \text{ bilirubin } (\mu\text{mol/L}) \times 0.66] + (\text{albumin } (\text{g/L}) \times -0.0852): \text{ grades } 1:2:3 = \leq -2.60:< -2.60 \text{ to } \leq -1.39 :> -1.39))$$

## Follow-up and diagnosis of HCC

HCC surveillance was performed with serum AFP tests and US every 6 to 12 months during and after antiviral treatment. When a focal lesion was identified in the liver on US, HCC was confirmed by imaging (CT or MRI) and/or biopsy. HCC was diagnosed histologically or based on imaging findings consistent with the diagnosis using at least two of the following methods: US, CT, MRI, and selective hepatic arteriography. The macroscopic classification,

histological differentiation, and architectural pattern were evaluated according to the classifications based on the general rules for clinical and pathological studies of primary liver cancer [37].

## Statistical analysis

We identified all new HCC cases diagnosed at least 180 days after the initiation of DAA treatment, as previously described. Continuous variables are described as the medians (interquartile ranges [IQRs]). Differences were compared using the Mann-Whitney U test and Wilcoxon matched-pairs signed-rank test. Receiver operating characteristic (ROC) curve analysis was used to determine the cut-off values for the factors associated with HCC occurrence. Multivariate Cox hazard regression analysis was performed to analyze the factors related to HCC occurrence after DAA treatment. We present confounder-adjusted hazard ratios (HRs) (adjusted for age, sex, AFP level, ALB level, FIB-4 score, presence of DM, PLT count, and ALBI score). Confounders with a threshold value lower than 0.05 in univariate analysis were selected a priori based on their known association with DAA treatment and HCC occurrence. The HCC development and survival rates were calculated using the Kaplan-Meier method, and the log-rank test was used to evaluate the differences between the curves. All statistical analyses were performed using Prism 6.0 software (GraphPad Software, Inc.) and JMP Pro 13.1 (SAS Institute Inc., Cary, NC). $P < 0.05$ was considered significant.

## Supporting information

**S1 Fig. Survival rate between hepatitis C patients without cirrhosis (NC) and those with compensated liver cirrhosis (LC).** (a) All survival rates of patients with compensated liver cirrhosis (red line) and patients without cirrhosis (blue line). (b) Non-liver-related survival rates of patients with compensated liver cirrhosis (red line) and patients without cirrhosis (blue line). (c) All survival rates of SVR patients with compensated liver cirrhosis (red line), non-SVR patients with compensated liver cirrhosis (pink line), SVR patients without cirrhosis (blue line), and non-SVR patients without cirrhosis (green line). (d) The non-liver-related survival rates of SVR patients with compensated liver cirrhosis (red line), non-SVR patients with compensated liver cirrhosis (pink line), SVR patients without cirrhosis (blue line), and non-SVR patients without cirrhosis (green line). The results were analyzed with the log-rank test.
(TIF)

**S2 Fig. Comparison of the cumulative hepatocellular carcinoma incidence and liver-related survival rate between hepatitis C patients without cirrhosis (NC) and those with compensated liver cirrhosis (LC).** (a) Cumulative incidence of hepatocellular carcinoma in SVR patients with compensated liver cirrhosis (red line), non-SVR patients with compensated liver cirrhosis (pink line), SVR patients without cirrhosis (blue line), and non-SVR patients without cirrhosis (green line). (b) The liver-related survival rate of SVR patients with compensated liver cirrhosis (red line), non-SVR patients with compensated liver cirrhosis (pink line), SVR patients without cirrhosis (blue line), and non-SVR patients without cirrhosis (green line). The results were analyzed using the log-rank test.
(TIF)

**S3 Fig. Comparison of the cumulative HCC incidence between HCV-positive patients without cirrhosis grouped according to the factors that might contribute to the development of HCC pre-SVR.** (a) Serum ALB level of <3.95 g/dl (red line); serum ALB level of ≥3.95 g/dl (blue line). (b) Serum AFP level of >6 ng/ml (red line); serum AFP level of ≤6 ng/

ml (blue line). (c) Presence of DM before DAA treatment (red line); absence of DM (blue line). (d) FIB-4 score of $\geq$3.25 (red line); FIB-4 score of <3.25 (blue line). The results were analyzed using the log-rank test.
(TIF)

**S4 Fig. Comparison of cumulative hepatocellular carcinoma (HCC) incidence between compensated cirrhosis patients with hepatitis C virus, grouped according to platelet count (PLT) or diabetes mellitus (DM).** (a) The PLT, <8.2 x10$^4$/μl (red line), $\geq$8.2 x10$^4$ (blue line). (b) The presence of DM before DAA treatment (red line), absence of DM (blue line). Results were analyzed by log-rank test.
(TIF)

**S1 Table. Characteristics of 1108 patients with hepatitis C virus (included 40 patients with non-SVR).**
(DOCX)

**S2 Table. Pretreatment factors associated with the development of hepatocellular carcinoma after DAA treatment in HCV-positive patients without cirrhosis (included 19 patients with non-SVR).**
(DOCX)

**S3 Table. Pretreatment factors associated with the development of HCC after DAA treatment in HCV-positive patients with compensated liver cirrhosis (includes 21 patients with non-SVR).**
(DOCX)

**S4 Table. Factors associated with the development of HCC after DAA treatment in HCV-positive patients without cirrhosis by Cox univariate analysis.**
(DOCX)

**S5 Table. Post-treatment factors associated with the development of hepatocellular carcinoma after DAA treatment in HCV-positive patients without cirrhosis.**
(DOCX)

**S6 Table. Post-treatment factors associated with the development of HCC after DAA treatment in HCV-positive patients without cirrhosis by Cox multivariate model.**
(DOCX)

**S7 Table. Post-treatment factors associated with the development of HCC after DAA treatment in HCV-positive patients with compensated liver cirrhosis.**
(DOCX)

**S8 Table. Post-treatment factors associated with the development of HCC after DAA treatment in HCV-positive patients with compensated liver cirrhosis based on the Cox multivariate model.**
(DOCX)

## Author Contributions

**Conceptualization:** Kazumichi Abe, Hiromasa Ohira.

**Data curation:** Kazumichi Abe.

**Formal analysis:** Kazumichi Abe.

**Investigation:** Kazumichi Abe, Hiroto Wakabayashi, Haruo Nakayama, Tomohiro Suzuki, Masahito Kuroda, Naoe Yoshida, Jun Tojo, Atsuko Kogure, Tsuyoshi Rai, Hironobu Saito, Shinji Mukai, Masashi Fujita, Manabu Hayashi, Atsushi Takahashi.

**Writing – original draft:** Kazumichi Abe.

**Writing – review & editing:** Kazumichi Abe, Hiromasa Ohira.

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
