## [Decision Letter · Decision Letter 0]

16 Sep 2020

PONE-D-20-23727

Factors associated with hepatocellular carcinoma occurrence after HCV eradication in patients without cirrhosis or with compensated cirrhosis

PLOS ONE

Dear Dr. Abe,

Thank you for submitting your manuscript to PLOS ONE. After careful consideration, we feel that it has merit but does not fully meet PLOS ONE’s publication criteria as it currently stands. Therefore, we invite you to submit a revised version of the manuscript that addresses the points raised during the review process.

We look forward to receiving your revised manuscript.

Kind regards,

Ming-Lung Yu, MD, PhD

Academic Editor

PLOS ONE

Journal Requirements:

2. In your ethics statement in the online submission form and in your manuscript text, please clarify whether the ethics committee specifically approved the study protocol.

" The funders had no role in study design, data collection and analysis, decision to publish, or preparation of the manuscript."

" The authors have declared that no competing interests exist."

We note that one or more of the authors are employed by a commercial company: Cosmos Clinic, Rai Clinic.

4.1. Please provide an amended Funding Statement declaring this commercial affiliation, as well as a statement regarding the Role of Funders in your study. If the funding organization did not play a role in the study design, data collection and analysis, decision to publish, or preparation of the manuscript and only provided financial support in the form of authors' salaries and/or research materials, please review your statements relating to the author contributions, and ensure you have specifically and accurately indicated the role(s) that these authors had in your study. You can update author roles in the Author Contributions section of the online submission form.

4.2. Please also provide an updated Competing Interests Statement declaring this commercial affiliation along with any other relevant declarations relating to employment, consultancy, patents, products in development, or marketed products, etc.  

Reviewers' comments:

Reviewer's Responses to Questions

**Comments to the Author**

1. Is the manuscript technically sound, and do the data support the conclusions?

Reviewer #1: Yes

Reviewer #2: Yes

2. Has the statistical analysis been performed appropriately and rigorously? 

Reviewer #1: Yes

Reviewer #2: Yes

3. Have the authors made all data underlying the findings in their manuscript fully available?

Reviewer #1: Yes

Reviewer #2: Yes

4. Is the manuscript presented in an intelligible fashion and written in standard English?

Reviewer #1: Yes

Reviewer #2: Yes

5. Review Comments to the Author

Reviewer #1: The manuscript entitled as “Factors associated with hepatocellular carcinoma occurrence after HCV eradication in patients without cirrhosis or with compensated cirrhosis” submitted by Abe et al. for possible publication in “PLOS ONE”. The authors investigated

the incidence of hepatocellular carcinoma (HCC) and factors related to HCC occurrence after direct-acting antiviral (DAA). The authors set the cohort as patients without cirrhosis (NC) or with compensated liver cirrhosis (LC). Multivariate analysis revealed serum albumin (ALB) level as independent factors affecting HCC occurrence for NC patients and the ALBI score, platelet count, and diabetes for LC patients. The authors concluded the factors related to HCC occurrence differed between NC and LC patients. Careful surveillance of post-SVR patients with these risk factors is needed.

The manuscript was well written, but some questions need to be clarified

Minor points:

1. The authors set the cohort as non-cirrhosis (NC) and cirrhosis (LC). The factors affecting HCC occurrence were quite different. Could the authors explain the differences of factors of multivariate analysis in NC and LC.

2. Could the authors offer the results of 40 patients without SVR. Could this group (without SVR) be included for analysis?

3. The HCC incidence was higher in NC group than in LC group at year 1 and 2? Could the authors explain?

4. Year 2 survival rate in LC group was higher than that in NC group? Could the authors explain?

5. 12 patients died in NC group, but only 7 patients died in LC group during study peroid? Could the authors identify causes of death and explain the difference?

Reviewer #2: The authors conducted a multicenter retrospective cohort study of 1068 patients without cirrhosis (NC) or with compensated liver cirrhosis (LC) who achieved a sustained virologic response (SVR). The objective was to explore the incidence and risk factors associated with HCC occurrence in non-cirrhotic and cirrhotic patients, respectively. The authors identified and concluded that independent factors affecting HCC occurrence were the serum albumin (ALB) level before SVR for NC patients and the ALBI score, platelet count, and diabetes before SVR for LC patients. In general, the manuscript was properly drafted. Nevertheless, some issues have to be clarified.

1. In the introduction, the authors stressed pretreatment factors to predict HCC. Certain reports regarding post-treatment factors are more determinant (Yu ML et al. Hepatology 2006.44(5):1086-97. Huang CF et al. Hepatol Int. 2018 Nov;12(6):544-551.)

2. DM or pre-DM has been a factor associated with HCC in SVR patients with mild liver disease (Huang CF et al. Medicine 2016 Jul;95(27):e4157. Hung CH et al. Int J Cancer 2011;128(10):2344-52) , which was in contrast to current report. The authors may have more discussion regarding the issue.

3. Why patients with decompensated liver cirrhosis were excluded in the current study

4. The authors have provided the data of survival rate. Did the author consider to adjust death as competing risk.

5. The definition of fatty liver is unknown and the proportion was low (6.9%). It is hard to believe that only one out of 188 cirrhotic patients had fatty liver (0.5%). Since the variable did not influence the occurrence of HCC, I would suggest to delete it in the analysis.

6. PLOS authors have the option to publish the peer review history of their article (what does this mean?). If published, this will include your full peer review and any attached files.

Reviewer #1: No

Reviewer #2: No

---

## [Author Response · Author response to Decision Letter 0]

4 Nov 2020

Response to Reviewer comments. 

Thank you for inviting us to submit a revised draft of our manuscript entitled, “Factors associated with hepatocellular carcinoma occurrence after HCV eradication in patients without cirrhosis or with compensated cirrhosis” to PLOS ONE. We also appreciate the time and effort you and each of the reviewers have dedicated to providing insightful feedback on ways to strengthen our paper. Thus, it is with great pleasure that we resubmit our article for further consideration. We have incorporated changes that reflect the detailed suggestions you have graciously provided. We also hope that our edits and the responses we provide below satisfactorily address all the issues and concerns you and the reviewers have noted.

Reviewer #1 

Minor points:

1. The authors set the cohort as non-cirrhosis (NC) and cirrhosis (LC). The factors affecting HCC occurrence were quite different. Could the authors explain the differences of factors of multivariate analysis in NC and LC.

RESPONSE: Thank you for providing these insights. In cases of liver cirrhosis, abnormal glucose metabolism, progression of liver fibrosis, and decreased hepatic reserve are associated, and factors related to these items are associated with hepatic carcinogenesis. In non-cirrhotic cases, there was no advanced liver fibrosis, and there was no problem with hepatic reserve. Thus, ALB, which was mentioned in previous reports, was extracted as a related factor. FIB-4, DM, and AFP were also extracted by univariate analysis but did not remain as independent factors in multivariate analysis.

2. Could the authors offer the results of 40 patients without SVR. Could this group (without SVR) be included for analysis?

RESPONSE: Thank you for your suggestion. We have included a new Supplemental Fig. S1-S2 and Supplemental Tables 1-3 to add the data of 1108 patients with hepatitis C virus including 40 patients without SVR (P7L14, Fig. S1-S2, Supplemental Tables 1-3). 

3. The HCC incidence was higher in NC group than in LC group at year 1 and 2? Could the authors explain?

RESPONSE: We agree with your assessment. We have reassessed the data of the HCC incidence and modified Fig. 2a and the manuscript (P3L7, P6L9, P13L14, P15L4). The 1-, 2-, 3-, and 4-year incidence rates of HCC were 0.7%, 1.1%, 1.8%, and 3.0%, respectively, for NC patients compared to 2.6%, 4.9%, 9.3%, and 11.5%, respectively, for LC patients (Fig. 2a). 

4. Year 2 survival rate in LC group was higher than that in NC group? Could the authors explain?

RESPONSE: We agree with your assessment. We have reassessed the data of survival rate and modified Fig. 2b. In the 880 NC patients with SVR, 2 patients died due to liver-related causes (HCC and hepatic ascites), and 10 patients died due to non-liver-related causes during the observation period. Of the 188 LC patients with SVR, 2 patients died due to HCC, 1 patient died due to liver failure, 1 patient died due to liver abscess, and 3 patients died due to non-liver-related causes during the observation period. The 1-, 2-, 3-, and 4-year liver-related survival rates were 99.9%, 99.8%, 99.8%, and 99.8%, respectively, for NC patients compared to 99.5%, 99.5%, 99.5%, and 98.5%, respectively, for LC patients (P3L7, P6L12, Fig. 2b).

5. 12 patients died in NC group, but only 7 patients died in LC group during study peroid? Could the authors identify causes of death and explain the difference?

RESPONSE: Thank you for your suggestion. In the 880 NC patients with SVR, 2 patients died due to liver-related causes (HCC and hepatic ascites), and 10 patients died due to non-liver-related causes during the observation period. Of the 10 patients with unrelated deaths, 2 patients died due to cholangiocarcinoma, 1 patient died due to gallbladder carcinoma, 1 patient died due to pancreatic carcinoma, 1 patient died due to renal cell carcinoma, 1 patient died due to malignant lymphoma, 1 patient died due to aortic dissection, and 1 patient died due to nontuberculous mycobacterial infection. In 2 patients, the cause of the death was unknown. In 188 LC patients with SVR, 2 patients died due to HCC, 1 patient died due to liver failure, 1 patient died due to liver abscess, and 3 patients died due to non-liver-related causes during the observation period. Of the 3 patients with non-related deaths, 1 patient died due to cerebral hemorrhage, and 2 patients died due to unknown cause (P6L12, Supplemental Fig. S1). 

Reviewer #2

1. In the introduction, the authors stressed pretreatment factors to predict HCC. Certain reports regarding post-treatment factors are more determinant (Yu ML et al. Hepatology 2006.44(5):1086-97. Huang CF et al. Hepatol Int. 2018 Nov;12(6):544-551.)

RESPONSE: Thank you for your suggestion. We have included new Supplemental Tables 5-8 to add data regarding post-treatment factors. We revised the sentence in the introduction (P4L11, P10L1, P11L9, Supplemental Tables 5-8).

2. DM or pre-DM has been a factor associated with HCC in SVR patients with mild liver disease (Huang CF et al. Medicine 2016 Jul;95(27):e4157. Hung CH et al. Int J Cancer 2011;128(10):2344-52), which was in contrast to current report. The authors may have more discussion regarding the issue.

RESPONSE: Thank you for your suggestion. The complication rate of DM was significantly higher in the HCC group than in the No HCC group, and DM was extracted as a related factor for the incidence of HCC in univariate analysis (Supplemental Table 4). However, DM was not extracted by multivariate analysis. In our study, non-cirrhotic cases contained a large amount of chronic hepatitis, and ALB was extracted by multivariate analysis rather than DM compared to the previously reported mild liver disease. We added this information to the discussion (P15L8, Supplemental Table 4).

3. Why patients with decompensated liver cirrhosis were excluded in the current study 

RESPONSE: Thank you for providing these insights. DAA treatment for decompensated cirrhosis began in Japan in February 2019. Therefore, it was not included in this study because the course of liver carcinogenesis cannot be tracked.

4. The authors have provided the data of survival rate. Did the author consider to adjust death as competing risk.

RESPONSE: We agree with your assessment. We have reassessed the data of survival rate and modified Fig. 2b. In the 880 NC patients with SVR, 2 patients died due to liver-related causes (HCC and hepatic ascites), and 10 patients died due to non-liver-related causes during the observation period. Of the 188 LC patients with SVR, 2 patients died due to HCC, 1 patient died due to liver failure, 1 patient died due to liver abscess, and 3 patients died due to non-liver-related causes during the observation period. The 1-, 2-, 3-, and 4-year liver-related survival rates were 99.9%, 99.8%, 99.8%, and 99.8%, respectively, for NC patients compared to 99.5%, 99.5%, 99.5%, and 98.5%, respectively, for LC patients (P3L7, P6L12, Fig. 2b).

5. The definition of fatty liver is unknown and the proportion was low (6.9%). It is hard to believe that only one out of 188 cirrhotic patients had fatty liver (0.5%). Since the variable did not influence the occurrence of HCC, I would suggest to delete it in the analysis.

RESPONSE: We agree with you. We have deleted the data regarding fatty liver.

---

## [Decision Letter · Decision Letter 1]

23 Nov 2020

Factors associated with hepatocellular carcinoma occurrence after HCV eradication in patients without cirrhosis or with compensated cirrhosis

PONE-D-20-23727R1

Dear Dr. Abe,

We’re pleased to inform you that your manuscript has been judged scientifically suitable for publication and will be formally accepted for publication once it meets all outstanding technical requirements.

Kind regards,

Ming-Lung Yu, MD, PhD

Academic Editor

PLOS ONE

Additional Editor Comments (optional):

Reviewers' comments:

Reviewer's Responses to Questions

**Comments to the Author**

1. If the authors have adequately addressed your comments raised in a previous round of review and you feel that this manuscript is now acceptable for publication, you may indicate that here to bypass the “Comments to the Author” section, enter your conflict of interest statement in the “Confidential to Editor” section, and submit your "Accept" recommendation.

Reviewer #1: All comments have been addressed

Reviewer #2: All comments have been addressed

2. Is the manuscript technically sound, and do the data support the conclusions?

Reviewer #1: Yes

Reviewer #2: Yes

3. Has the statistical analysis been performed appropriately and rigorously? 

Reviewer #1: Yes

Reviewer #2: Yes

4. Have the authors made all data underlying the findings in their manuscript fully available?

Reviewer #1: Yes

Reviewer #2: Yes

5. Is the manuscript presented in an intelligible fashion and written in standard English?

Reviewer #1: Yes

Reviewer #2: Yes

6. Review Comments to the Author

Reviewer #1: The authors have done appropriate revisions according to my questions. I don't have more questions.

Reviewer #2: The present study aimed to investigate the incidence of hepatocellular carcinoma (HCC) and factors related to HCC occurrence after direct-acting antiviral (DAA) treatment in the Fukushima Liver Academic Group (FLAG). I have no further comment upon the revised manuscript

7. PLOS authors have the option to publish the peer review history of their article (what does this mean?). If published, this will include your full peer review and any attached files.

Reviewer #1: No

Reviewer #2: No

---

## [Editor Report · Acceptance letter]

27 Nov 2020

PONE-D-20-23727R1 

Factors associated with hepatocellular carcinoma occurrence after HCV eradication in patients without cirrhosis or with compensated cirrhosis 

Dear Dr. Abe:

I'm pleased to inform you that your manuscript has been deemed suitable for publication in PLOS ONE. Congratulations! Your manuscript is now with our production department. 

Kind regards, 

on behalf of

Dr. Ming-Lung Yu 

Academic Editor

PLOS ONE